# Synchronous Front-Face Fluorescence Spectra: A Review of Milk Fluorophores

**DOI:** 10.3390/foods13050812

**Published:** 2024-03-06

**Authors:** Paulina Freire, Anna Zamora, Manuel Castillo

**Affiliations:** 1Centre d’Innovació, Recerca i Transferència en Tecnologia dels Aliments (CIRTTA), Department of Animal and Food Sciences, Universitat Autònoma de Barcelona, Bellaterra, (Cerdanyola del Vallès), 08193 Barcelona, Spain; paulinaf@mail.fresnostate.edu (P.F.); manuel.castillo@uab.es (M.C.); 2Department of Food Science and Nutrition, California State University, Fresno, 5300 N CampusDrive M/S FF17, Fresno, CA 93740, USA

**Keywords:** fluorophores, excitation emission matrix, skim milk, synchronous fluorescence spectroscopy, front-face fluorescence

## Abstract

Milk is subjected to different industrial processes, provoking significant physicochemical modifications that impact milk’s functional properties. As a rapid and in-line method, front-face fluorescence can be used to characterize milk instead of conventional analytical tests. However, when applying fluorescence spectroscopy for any application, it is not always necessary to determine which compound is responsible for each fluorescent response. In complex matrixes such as milk where several variables are interdependent, the unique identification of compounds can be challenging. Thus, few efforts have been made on the chemical characterization of milk’ fluorescent spectrum and the current information is dispersed. This review aims to organize research findings by dividing the milk spectra into areas and concatenating each area with at least one fluorophore. Designations are discussed by providing specific information on the fluorescent properties of each compound. In addition, a summary table of all fluorophores and references cited in this work by area is provided. This review provides a solid foundation for further research and could serve as a central reference.

## 1. Introduction

Milk is a complex matrix of water, fat, protein, lactose, minerals, traces of pigments, enzymes, and vitamins. Milk composition may vary considerably between breeds and even between individuals of the same breed, as well as seasonal changes, age, stage of lactation, and feeding [1]. In addition, milk is subjected to different processes, such as homogenization or heat treatments, which impact milk functional properties such as emulsification, coagulation, foaming, and film formation [2,3].

Milk characterization, which implies various test analyses, is necessary for dairy processing to ensure milk functional quality. Most of them are time- and material-consuming, not feasible for in-, on- or at-line monitoring, and require a qualified workforce [4]. On the contrary, front-face fluorescence spectroscopy has shown great potential to assess dairy products and displays various advantages over other methods since it is a fast, sensitive, and reliable technique that does not destroy the sample, making it suitable for in-, on- or at-line process control [5]. Indeed, it has been demonstrated that it can distinguish between different levels of heat treatment [6,7], identify the geographical origin of the milk [8], differentiate between homogenized and non-homogenized milk [9], predict functional properties of whey proteins [10], predict particle size of caseins [11], predict the concentration of lactulose [12], native whey protein, aggregate whey protein [11], hydroxymethylfurfural, sulfhydryl groups [13], riboflavin, ascorbic acid [14], furosine [15], as well as, model the kinetics of retinol loss during thermal processing [16], and predict storage time in infant milk formula [17]. 

Some milk compounds are naturally fluorescent, known as fluorophores, where each fluorophore has its specific excitation and emission profile [18] and could show slight changes in protein and lipid structure [19,20], providing milk quality information. Various studies presented the spectrum behavior of each fluorophore, but separately. It means that milk’s fluorophores were scanned by setting a specific excitation wavelength and displaying an emission spectrum [6,8,9,17,21] or using a range of different excitations, where a single emission was recorded [3,9,22]. However, several measures should be performed on multi-component samples, such as milk, to assess more than one fluorophore. This requires knowing the optimum emission or excitation wavelength for each fluorophore previously, which will change depending on the milk’s conditions.

Synchronous fluorescence is an excellent way to collect fluorescence from all fluorophores at once. It scans both a range of excitation and emission simultaneously with a fixed interval between the excitation and emission wavelengths [23,24]. 

The combination of synchronous spectroscopy with NIR spectroscopy or fluorescence enhancers can improve sensitivity and specificity in detecting trace substances in complex matrices [25,26]. Synchronous fluorescence fingerprints could act as the identity card of food products. The non-specific fingerprinting approach relies on the implementation of instrumental methods to obtain a multivariate description of the chemical composition of the sample and on the statistical approaches to extract meaningful information, known as chemometric models. Machine learning approaches, like support vector machines or artificial neural networks, can complement traditional chemometric methods in spectral data analysis and wavelength peak identification [27]. As a result, it can provide broader information, and no specific wavelength is necessary. Most synchronous fluorescence studies on milk have focused on a specific application. For example, the synchronous fluorescence method has been used to identify between different milk species [28], detect the presence of reconstituted milk in raw milk and in pasteurized milk [29], quantify protein content [30], classify the types of protein leaks in permeate streams [30] and identify the impact of milk heat treatment on coagulation [31]. Some studies have attempted to assign the observed peaks to a compound by referring to other researches [32,33]. Indeed, to the best of our knowledge, no study has systematically assigned or related each peak, that shows up on whole synchronous fluorescence spectra of milk, to a specific fluorophore. Therefore, this study aims to organize literature information by dividing the spectra into areas and concatenating each area with at least one compound based on the fluorescent properties.

## 2. General Fluorophores Features

A fluorophore is a type of molecule that can re-emit light of other frequencies after being stimulated by another light excitation. Fluorophore molecules absorb at higher and emit lower energy radiation where fluorophore’s electrons perform the energy transition. The general principle can be illustrated by a Jablonski diagram (Figure 1), in which an electron’s energy transition between energy states is presented as vertical lines, but not all transitions are allowed. The energy involved in an electron’s transition is the energy difference between the starting energy level and the final level, according to the equation: E=hυ; where h is the Planck constant and υ is the light frequency. 

A fluorophore can absorb energy by photons of light, which should have higher energy than the ground state (S_0_) to promote an electron at some higher vibrational level of either S_1_ or S_2_. This excited state is unstable, and thus, a vibrational relaxation or internal conversion occurs, where there is a first loss of energy but no light emission. This energy is liberated as vibrational energy, which is converted into heat absorbed by closed molecules. A final relaxation occurs when the electron returns to the ground state. During this transition, light is emitted. Therefore, the energy released is always smaller than the energy acquired by the electron in the first place. Since high-wavelength radiation is less energetic than short-wavelength radiation, the emission wavelength is always greater than the excitation one. The wavelength difference between the peak of the excitation and that of the emission is known as Stokes Shift (Figure 2 [5,34]).

A fluorophore can absorb and emit photons of light with different levels over the distribution of wavelengths, corresponding to different vibrational transitions, displayed as spectra. In general, the chemical configuration of fluorophores contains one or more rings, usually aromatic structures, and sometimes certain carbonyl groups with aliphatic chains [35]. Fluorophores are classified into two general groups: intrinsic and extrinsic [34]. The intrinsic group is molecules that occur naturally, while the extrinsic group is synthetic or modified molecules that are added to display fluorescence. Variations in temperature and environment, such as pH, solute concentration, solvent polarity, and color, may modify the fluorophore fluorescence features. For example, when the environment is less polar, the emission peak shifts to shorter wavelengths (blue-shift) and its intensity increases, but the change is tiny in a non-polar environment [36]. Consequently, a fluorophore is sensitive to the environment and could be used as a marker of surroundings. In addition, the fluorescence intensity response of a fluorophore can decrease due to a phenomenon called fluorescence quenching, which is when an excited fluorophore transfers energy to another molecule in a solution [37]. Thus, fluorescence offers several advantages for the characterization of food products due to its sensitivity to molecular interactions, chemical reactions, and changes in the environment of molecules.

## 3. Fluorescence Measurements

### 3.1. Fluorescence Geometry

Fluorescence measurements can be performed using two different geometries: the right angle and the front-face fluorescence. Regarding the first geometry, the excitation light travels through the sample and the detector is positioned at a right angle to the center of the sample, as shown in Figure 3 Therefore, it can be applied only in transparent samples, and a dilution should be performed on turbid samples. Alternatively, the second geometry requires the sample be placed with its surface oriented between 30° to 60° with respect to the axis of inflection (Figure 2). In this manner, the emission and the excitation occur on the same cuvette face, which means that the front-face can be used for opaque liquids, such as milk or solid samples. Consequently, there is no need for sample manipulation, and it is suitable for in/on/at-line applications [34,36]. This configuration permits the implementation of this technology into miniaturized instrumentation, compact spectral sensors, and handheld systems or portable devices.

Most of the studies reporting the use of fluorescence in milk have been performed by using the so-called classical front-face fluorescence spectroscopy with wavelength ranges detecting molecules such as tryptophan and Maillard compounds. Many studies focused on assessing the potential of front-face fluorescence spectroscopy to characterize and/or differentiate heat treated and commercial milk [7,9,11,13,15,22,38]. Front-face fluorescence spectroscopy has also been assessed to identify milk geographic origin [8], feeding systems [39], differentiate between Sicilo–Sarde and Comisana Ewe’s Milk [40], and to detect adulteration of goat milk powder with bovine milk powder [41].

### 3.2. Fluorescence Measurements

There are different ways to collect fluorescent measurements from a sample to display its spectra (emission spectrum, excitation spectrum, and total excitation–emission spectrum). The most conventional is when the excitation monochromator is set at a single wavelength, and the emission spectrum of a sample is scanned. Usually, the molecule is excited at its absorption maximum (λex_max_). The inverse is another way where a range of different excitations is used, and a single emission is recorded. In this case, the fluorescence measurement collects the emission maximum (λem_max_). Only one excitation and one emission spectrum are usually enough to characterize a pure compound. However, samples with multiple fluorophores, such as milk, may require scanning multiple times at different excitations and emissions. It also implies that previous knowledge of each fluorophore’s appropriate excitation/emission.

The Excitation–Emission Matrix (EEM) way increases the information, where a range of emission spectra is scanned at different excitation wavelengths, and three-dimensional data is obtained (excitation, emission, and intensity). In this method, some information needs to be removed before data analysis. For example: (1) when the excitation wavelength is higher than the emission. It is because there is no fluorescent response. (2) when the excitation and emission wavelengths are equal. It constitutes the scattered light, known as 1st order light. According to measurement principles, the second order light should also be deleted. This happens when the emission wavelength is twice that of the excitation wavelength. (3) when values are under 230 nm and over 810 nm for excitation and emission, respectively. It is highly recommended to be removed because they are regions with low sensibility. Thus, this method requires data preprocessing since not all information is fluorescent response [42,43,44].

On the contrary, the excitations and emissions are simultaneously scanned in synchronous measurements with a fixed interval between excitation and emission wavelengths, named delta (Δλ). This method allows spectral simplification and always maintains the excitation smaller than the emission wavelength. In other words, the light scattered is excluded by itself [23,42,45]. Regarding the use of front-face synchronous fluorescence spectroscopy, studies have focused on milk heat treatment [46,47,48,49], and milk quality by detecting species adulteration [26,28,33,50].

## 4. Synchronous Front-Face Fluorescence of Milk

Milk contains various natural fluorescent compounds, some of which are essential for nutritional value and technological functionality. These compounds include amino acids, cofactors, and vitamins, among others, which serve as intrinsic fluorophores. This section aims to organize the literature information on milk fluorophores based on the front-face fluorescence response of reconstituted skimmed cow milk powder (“low heated”, spray-dried, pH = 6.5, solubility = 99%, WPNI ≥ 7 mg·g^−1^ and 800 cfu·g^−1^ milk; Chr. Hansen SL, Jernholmen, Denmark). From the heatmap of milk excitation-emission matrix spectra (250–550 nm and 300–750 nm, respectively), ten areas of interest were detected (Figure 4), and each area was concatenated with at least one fluorophore. A summary table of the potential fluorophores is provided at the end of the discussion.

### 4.1. Area 1 (Exc. 250–280 nm/Em. 420–450 nm)

The literature does not provide information on the fluorescence response of this area in cow milk. However, fluorophores that emit over 390 nm have been attributed to carotenoids and retinol (vitamin A) in yogurt and cheese samples [51]. Carotenoids are synthesized naturally in plants but not in animals. Carotenoids are a group of pigments that are naturally found in plants, algae, and some bacteria. They are responsible for the yellow, orange, and red colors of fruits and vegetables, and are known to have numerous health benefits. One of the lesser-known sources of carotenoids is milk, particularly cow milk. The most abundant carotenoid is β-carotene, also known as provitamin A, and it can form two retinol molecules [52]. Therefore, β-carotene is metabolized easily into retinol. Both are fluorescent in milk and are considered relevant molecules in food due to their nutritional value [50]. Carotenoids and retinol are fat soluble; they can be found mainly in the fat globule of milk [53], but it has been demonstrated that retinol forms water-soluble complexes with whey proteins such as α-Lactalbumin and β-Lactoglobulin [54,55]. In addition, although reconstituted skimmed milk was used, traces of retinol may still be in the reconstituted milk since the fat content of milk powder was 1.25%. Another feature of carotenoids and retinol is that they are sensitive, especially in the presence of light and heat [52]. Moreover, the absorption of light by β-carotene may affect the measured fluorescence of retinol [20]. In various studies, by setting the emission wavelength to ~410 nm, a peak at ~300 nm was observed in the excitation spectra, attributed to retinol [3,9,22].

### 4.2. Areas 2 and 3 (Exc. 270–280 nm/Em. 323–355 nm and Exc. 280–298 nm/Em. 323–355 nm)

These two areas are small, with only a 10 nm range in excitation. Notice that they differ in the excitation wavelength since the emission wavelengths are identical. Some of the intrinsic fluorescence of milk proteins is due to three aromatic amino acids: phenylalanine (Phe), tyrosine (Tyr), and tryptophan (Trp), which are generally used as markers to assess protein. They emit from 250 to 440 nm, but their emission wavelengths overlap [37], making it hard to evaluate them separately. The quantum yield of Phe is too low to be detected (0.02) and can be observed only in the absence of Tyr and Trp [56]. Although the fluorescence intensity of Tyr is higher than Trp in solution, these two amino acids’ quantum yields are almost identical: 0.14 and 0.13, respectively. It occurs because the Trp emission spectrum is wider than Tyr, which gives the appearance of a lower quantum yield [37]. Several studies in milk reported the emission fluorescence spectrum of Trp at an excitation of 290 nm and obtained a peak intensity of around 340 nm [6,8,9,17]. It has been demonstrated that conformational changes in protein could be studied mainly by changes in the fluorescence intensity of Trp but also of Tyr [57]. In addition, Trp has been used in milk to identify thermal treatments [6], distinguish between species [51] and establish changes in casein micelle during pH changes [21]. For example, Ayala et al. [6] found that the fluorescence intensity of Trp decreased and observed a shift toward larger wavelengths, called redshift, which was observed when there was an increase in the intensity of thermal treatments in skim milk by assessing the wavelengths: 290/340 nm (excitation/emission) (Figure 5). The synchronous response of yogurt from different milk species showed a standout peak around 280–300 nm and 320–350 nm in excitation and emission, respectively. This peak distinguished yogurt produced with buffalo milk from those of cow, goat, and ewe, and was attributed to Trp [51]. In contrast, the fluorescence emission of Trp (305–400 nm) was measured during the acidification of reconstituted milk, and a redshift was detected on the emission maximum [21]. Tryptophan is very sensitive to the environment. Its fluorescent spectrum is well characterized and widely used to assess milk. Just to cite a few examples: it has been observed that tryptophan spectra could be used to distinguish between different levels of heat treatment [6,7], to identify the geographical origin of milk [5,33], to differentiate between homogenized and non-homogenized milk [9], to check milk authenticity [58], to investigate protein interactions [59] and to predict storage time in infant milk formula [17].

According to Andersen and Mortensen [20], emissions between 305 and 400 nm are spectroscopic parameters widely used to explain and detect protein structure changes such as: conformational transitions, associations, and denaturation. When Tyr is in proteins, its fluorescence is lower than Trp even if Tyr residues are equal to or greater than Trp. An explanation for this could be the position of Tyr in a tertiary structure, resulting in Tyr residues hiding. Another possibility could be that Tyr transfers energy to Trp during the excitation, inducing a quenching of Tyr [34]. A study recording 3D fluorescence spectra of pure grade Tyr, at the excitation range of 200–450 nm and emission range of 200–500 nm, found two peaks with the strongest intensity was at 280/305 nm excitation/emission [60]. In addition, Christensen et al. [18] suggested wavelengths of 276 and 302 nm for Tyr excitation and emission, respectively. Therefore, Tyr was expected to emit at a slightly shorter wavelength than Trp. However, Murillo et al. [61] observed two peaks with almost the same emission ~331 nm in spectra of different whey samples (excitation/emission ranges of 220–320/260–435 nm). A chromatographic study was used to prove that whey fluorescence in these areas was due to the presence of Tyr and Trp. Consequently, area 2, with average excitation/emission wavelengths of 278/338 nm, could be attributed to Tyr. Meanwhile, area 3, with average excitation/emission wavelengths of 285/337 nm, would be associated with Trp.

### 4.3. Area 4 (Exc. 320–350 nm/Em. 390–445 nm)

Ma and Amamcharla [30] analyzed the fluorescence spectra of milk and whey permeate powders recording the emission spectra from 300 to 500 nm and the corresponding excitation wavelengths from 220 to 360 nm. They attributed the fluorescence at 310–350 and 380–430 nm excitation/emission to Maillard products, which are a series of compounds that form through chemical chain reactions between amino groups of amino acids and reducing sugars. One consequence of these glycation reactions in proteins is the formation of covalently cross-linked aggregates, and some of them are fluorescent [62]. For the sake of simplicity, the Maillard reaction can be divided into three main stages: early, advanced, and final. These stages are interrelated and are affected by the conditions [63]. In milk, Maillard reactions occur during heat treatment or storage [64]. Pentosidine is a cross-link molecule formed between a modified lysine residue and arginine [65]. It is considered as a representative of the advanced glycation end products (AGEs) and is a fluorescent molecule [66]. It excites at 335 nm and emits at 385 nm when isolated and in its native form [65]. These fluorescence values are very close to those found in the present study. It should be remembered and considered that the milk used is from low-heat skim milk powder, which means that it had previously been subjected to a heat treatment during the dehydration process. Moreover, pentosidine has been detected in milk samples such as UHT, sterilized, evaporated, and powdered [67]. Therefore, it makes sense to attribute area 4 to pentosidine.

Another compound that could be related to the fluorescence obtained in area 4 is dityrosine, which is a specific product of protein oxidation. This fluorescent molecule is proposed as a marker of oxidative conditions. Solution of pure dityrosine exhibited emission and excitation spectra with one peak, around 315 nm for the emission when excited at 400 nm [68].

The values of excitation and emission observed in area 4 were considerably similar to those detected by Blecker et al. [31], where synchronous measurements were performed on renneted skim raw milk and presented a peak around 333/393 nm (excitation/emission). This peak was attributed to pyridoxine (vitamin B6), a water-soluble fluorescent vitamin presents in small amounts in cow milk [69]. Christensen et al. [18] proposed the same vitamin as a characteristic fluorophore of food samples, with similar wavelength values, around 328/393 nm. Another vitamin that could contribute with its fluorescence to area 4 is retinol. Its maximum emission wavelength is located at around 412 nm in milk samples when the excitation is set at 321 nm [9]. However, Ali et al. [49] suggests that the emission peak at 410 nm indirectly corresponds to retinol in buffalo milk. As mentioned for area 1, retinol is a fat-soluble vitamin, but it can form hydrophilic compounds when retinol binds to whey proteins.

As set out above, many molecules could be responsible for the fluorescence of area 4 such as Maillard compounds, i.e., pentosidine, dityrosine, vitamin B6 or retinol.

### 4.4. Area 5 (Exc. 350–380 nm/Em. 440–470 nm)

Area 5 could also be attributed to Maillard compounds. Indeed, some fluorescent products from the Maillard reaction can be observed in different food systems with excitations and emissions of 340–370 and 420–470 nm, respectively [62]. Obayashi et al. [70] suggested that some AGEs could be excited from 340 to 370 nm and emitted from 420 to 440 nm wavelength. In another study, with camel milk, the emission spectra of Maillard compounds (380–680 nm) were obtained by exciting at 360 nm, and presented three peaks at 450, 480, and 510 nm [22]. It must be noticed that peak wavelengths of 360/450 nm (excitation/emission) are very similar to those of area 5. The molecular structure of many AGEs has not yet been specified. In parallel, medical studies on human skin ageing used excitation and emission wavelengths of 370 and 440 nm to assess the accumulation of AGEs [71,72]. In addition, the same excitation/emission wavelengths were used by Wu et al. [73] to measure the fluorescence associated with AGEs produced after incubating D-glucose and bovine serum albumin (BSA). Specifically, there is a fluorescent compound with excitation/emission wavelengths of 366/440 nm called pentodilysine, an AGE product resulting from the interaction between pentoses or ascorbic acid and lysine residues, all of them available in milk [74].

Another fluorophore that excites and emits close to area 5 is NADH, an enzyme cofactor found in milk. The excitation/emission of NADH in water is 340/460 nm, but upon binding of NADH to protein, its fluorescence may change depending on the protein [37]. Its emission fluorescent response was assessed in camel milk after thermal treatments by setting the excitation wavelength at 340 nm [22]. A peak located at 450 nm in the emission spectra, whose intensity decreased as the heat treatment increased, was observed and was attributed to NADH. In the same line, Kulmyrzaev et al. [38] used front-face fluorescence at an excitation of 360 nm to characterize milk after different thermal treatments. These authors found two peaks and suggested that the smallest one, at wavelengths of 360/460 nm (excitation/emission), could be NADH. In conclusion, area 5 of milk identified in the present research could be attributed to either Maillard compounds, specifically pentodilysine, or to NADH.

### 4.5. Areas 6, 7, and 8 (Exc. 368–380 nm/Em. 415–430 nm; Exc. 370–400 nm/Em. 505–535 nm; Exc. 390–410 nm/Em. 410–430 nm)

Since AGEs can be excited from 340 up to 370 nm emitting from 420 to 440 nm [70], area 6 could also be related to these molecules. Indeed, Birlouez-Aragon et al. [75] used 350/440 nm (excitation/emission) in the milk fraction soluble at pH 4.6 to assess the global formation of Maillard compounds after different heat treatment processes were applied to milk samples. In addition, fluorescence in infant formulas was measured during storage by setting 415 nm in emission and the excitation from 200 to 400 nm, to assess AGEs compounds [76]. Three peaks in the excitation spectra were observed: 270, 315/325 and 350 nm, where the highest wavelength (350 nm) is near the values obtained for area 6. However, the authors did not specify the compound which was responsible for the fluorescence. A molecule that can fit in this area is pyrropyridine since its excitation/emission is 370/455 nm [77]. It is an AGE product resulting from the reaction of 3 deoxyglucosone and lysine, both compounds available in milk. Although the excitation wavelength value is similar to that of area 6, the emission from pyrropyridine is much higher. Maillard compounds have not been fully established, and fluorescent products even less, mainly due to the great number of compounds that can be formed, and the complexity of the pathways involved. Consequently, a deeper investigation would be necessary to relate this area to a specific compound.

According to Yang et al. [78], riboflavin, also known as vitamin B2, excites at 370 nm and emits at 525 nm, which fits with area 7. However, riboflavin has three well defined peaks located at around 270/525 nm, 370/525 nm and 450/525 nm, which could correspond to other areas of the present study thus area 7 will be discussed in Section 4.7.

Area 8 was also attributed to riboflavin, which could correspond to the other peak of this compound. Its discussion can be found in Section 4.7, together with areas 7 and 10.

### 4.6. Area 9 (Exc. 410–430 nm/Em. 410–450 nm)

It must be noticed that this area has a difference of a few nanometers between excitation and emission. The emission area from 400 to 500 nm is complex because the fluorescence observed here may be from compounds degraded by light, such as: riboflavin or retinol, and Maillard compounds [20]. The compound, which possesses fluorescent attributes, is closest to area 9 is lumichrome, whose fluorescence response in the emission range is around 444–479 nm [79]. Lumichrome occurs naturally in milk exposed to light since it is a photo-chemical degradation product from riboflavin [80]. Hence, lumichrome is in the group of the flavins. Most of the flavins excite in similar values than riboflavin. Andersen and Mortensen [20] point out that the excitation maximum of lumichrome is similar as the excitation maximum of riboflavin, and thus, about 450 nm. Koziol [81] reported that water solution of lumichrome excites at 219, 261 and 354 nm, which means that area 1 and 5 may be also involved while in 2 N NaOH solution excites at 224, 266, 340, and 432 nm, where area 9 may be also implicated.

### 4.7. Area 10 (Exc. 425–480 nm/Em. 515–540 nm)

After analyzing the fluorescent behavior of areas 7, 8 and 10, it was hypothesized that the fluorescent response is due to riboflavin, which is known for its strong fluorescence in dairy products [20]. A study on the fluorescence spectra of riboflavin in aqueous solution scanned at an excitation wavelength from 200 to 550 nm and an emission range of 450–650 nm, showed three excitation peaks: 270, 370 and 450 nm at the same emission wavelength, 525 nm [78]. This emission wavelength matched with the emission values of areas 7 and 10, which were 505–535 nm and 515–540 nm, respectively. Notice that these two areas had the highest emission values compared with the other areas studied. The maximum intensities were observed at about 380/520 nm and 463/528 nm (excitation/emission) for area 7 and 10, respectively. Alvarado [13] and Ayala [15] explored the riboflavin by exciting at 267, 370 and 450 nm, and emission peaks were found at around 535, 507 and 522 nm, respectively. Consequently, area 7 could be attributed to the second emission peak of riboflavin and area 10 to the third one. Bhattacharjee et al. [82] present the spectra of milk and pure riboflavin excited at 420 nm and emitted at 520 nm, showing the third peak of the riboflavin belonging to area 10. Regarding the first peak, it is essential to highlight that in this study, the first peak of riboflavin, located around 267/535 nm (excitation/emission), was out of the scanning range since the Delta stop was set at 220 nm. Therefore, at 267 nm of excitation, the maximum emission scanned was 487 nm.

As was mentioned before, in area 8 a saturation occurred. However, since it was a small area, it could be characterized. Karoui et al. [8], with an excitation wavelength of ~380 nm, reported one peak in milk emission spectrum at 520 nm and a smaller one at ~420 nm, which was present in milk samples produced in lowlands but absent in milk from mid-mountain and mountain areas. Therefore, the peak at 380/420 nm (excitation/emission), attributed to riboflavin, could fit with area 8. Alvarado [13] showed the emission spectra of the riboflavin from heat-treated skimmed milk samples, but the excitation was lower (370 nm) than that of Karoui et al. [8]. As a result, two peaks were also observed in the emission spectrum, one at about 485 nm and the other at 522 nm (Figure 6).

Riboflavin contains a nitrogen base, called flavin group, which is fluorescent. The same group is present in the coenzyme flavin adenine dinucleotide (FADH_2_). This coenzyme is water-soluble and photostable. In addition, it has been demonstrated that the fluorescent intensity of FADH_2_ increases when it is in a disintegration state [81]. Only the reduced form is highly fluorescent and occurs naturally in milk. As for riboflavin, FADH_2_ emits at high wavelengths. According to Albani [34], FADH_2_ excites at ~450 nm and emits at 515 nm. These values fit also with the fluorescence response of area 10. In another study, milk heat-treated samples were characterized by recording the excitation spectra setting the emission wavelength at 518 nm. In this study, two peaks were observed where the second peak, located at ~445 nm, was associated with the FADH_2_ molecule [38]. Therefore, the fluorescence observed in area 10 could be attributed either to riboflavin or to FADH_2_ coenzyme.

### 4.8. Other Compounds

Other intrinsic fluorophores that can be found in milk are chlorophyll and chlorophyll metabolites. When cows feed on grass, the milk they produce has a higher concentration of chlorophyll. Chlorophyll compounds are present in almost all milk products in variable amounts. Their emission peaks are found between 600 and 700 nm with excitation lights of 380 nm [83] or 420 nm [82]. It has been demonstrated that the fluorescence of chlorophyll can help to discriminate the milk of grass-fed cows from that of cows fed with conventional grain [82]. A specific type of chlorophyll metabolite called pheophorbide can also be detected by fluorescence spectroscopy. This metabolite is known to exhibit a peak at around 675 nm, along with a smaller peak at around 720 nm. Based on these findings conducted by Bhattacharjee et al. [82], the chlorophyll metabolite content significantly increases when cows consume grass as their main source of food.

Below, Table 1 summarizes the potential fluorophores for each area.

## 5. Conclusions

Considering that milk is a complex product containing many fluorescent molecules, it is convenient to evaluate the synchronous spectrum by areas. Each area was concatenated with at least one compound through literature review facilitating the characterization of the whole spectrum of milk.

This review revealed that only a limited number of investigations have been conducted to chemically verify, or even indirectly verify, the compound in question. As a result, a substantial portion of the available knowledge is based on the analysis of fluorescent responses in other systems, such as other food or pharmaceutical matrix.

This study, by compiling all the information on milk fluorescence response in just one work, lays the foundation for any study on the synchronous fluorescence of milk and could serves as a central reference. However, it must be born in mind that, for many fluorophores, confirmation through quantification is missing.

## Figures and Tables

**Figure 1 foods-13-00812-f001:**
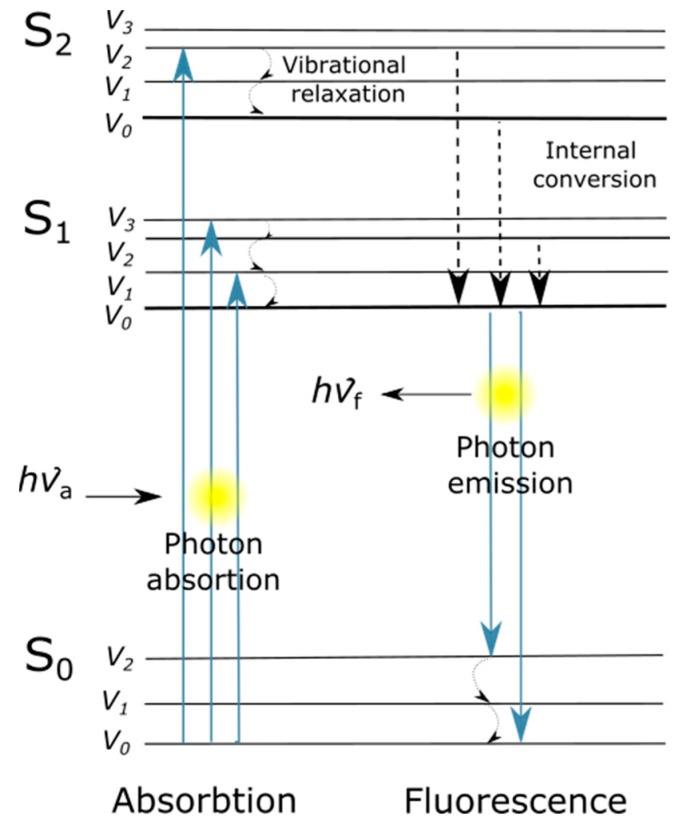
A simplified Jablonski diagram to illustrate fluorescence process.

**Figure 2 foods-13-00812-f002:**
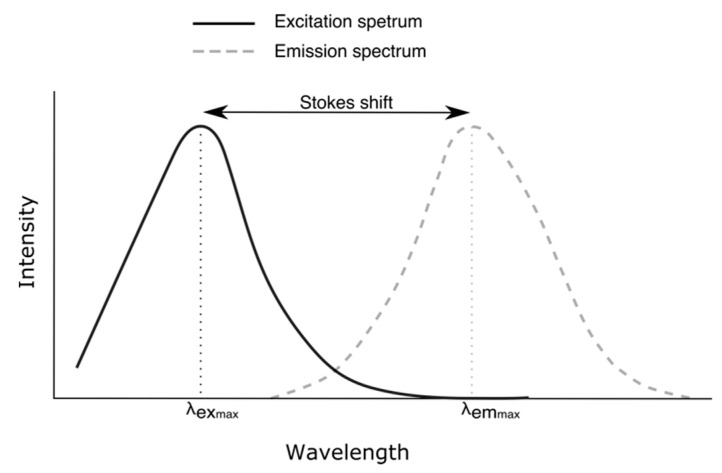
Excitation and emission spectra.

**Figure 3 foods-13-00812-f003:**
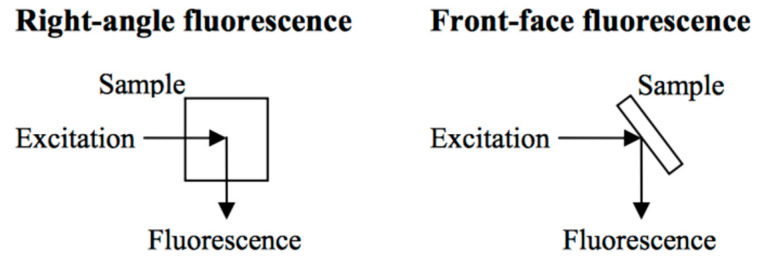
Principles of right-angle and front-face fluorescence configurations.

**Figure 4 foods-13-00812-f004:**
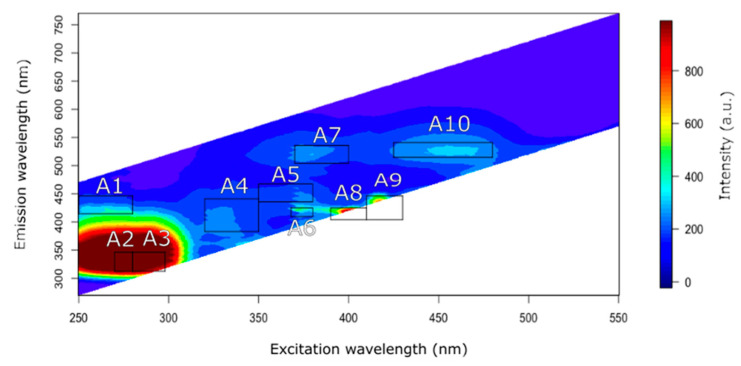
Areas of interest established on heatmap fluorescence of reconstituted skimmed milk powder (“low heated”, spray-dried, pH = 6.5, solubility = 99%, WPNI ≥ 7 mg·g^−1^ and 800 cfu·g^−1^ milk; Chr. Hansen SL, Jernholmen, Denmark). Milk spectra were acquired at 20 °C on a fluorescence spectrophotometer (Cary Eclipse Fluorescence Spectrophotometer, Agilent Technologies, Santa Clara, CA, USA) equipped with 15 W lamp “press Xenon lamp” and a “front-face” geometry accessory at 35° (Solid Sample Holder Accessory and Cuvette Kit, Agilent Technologies). The excitation was scanned from 250 to 550 nm at 600 volts with 1 nm intervals and an initial Delta of 20 nm, a Delta increment of 10 nm from the corresponding excitation wavelength and a Delta stop of 220 nm (excitation and emission slit widths of 5 nm); A1–A10: areas 1 to 10.

**Figure 5 foods-13-00812-f005:**
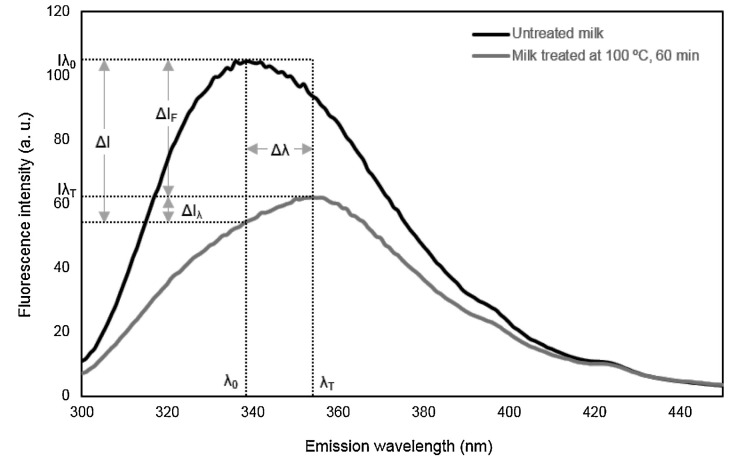
Schematic figure showing the decomposition of the decrease in total fluorescence intensity of tryptophan (ΔI) into two fractions, one associated with the change of the peak height (ΔI_F_) and another with the displacement of the emission maximum (ΔI_λ_). Change in the peak wavelength (Δλ) can be seen as a shift from λ_0_ to λ_T_ with intensities shown as Iλ_0_ and Iλ_T_. Reprinted with permission from Ref. [6]. Copyright 2020, Elsevier.

**Figure 6 foods-13-00812-f006:**
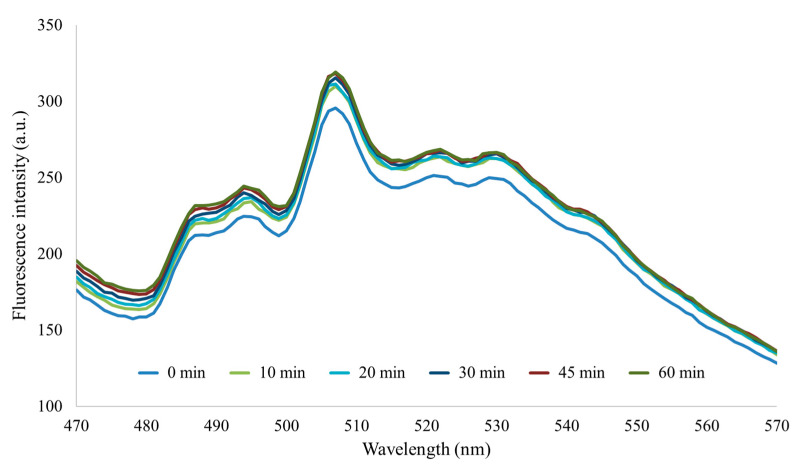
Emission spectra excited at 370 nm of reconstituted skim milk powder heat treated at 80 °C at six holding times.Reprinted with permission from Ref. [13]. Copyright 2016, Universitat Autònoma de Barcelona.

**Table 1 foods-13-00812-t001:** Summary table of the fluorophores and references cited in this work by area.

Area	Fluorophore	Excitation Wavelength (nm)	Emission Wavelength (nm)	Reference
A1	Carotenoids	>390	300–550	[51]
	Vit A	321 shoulders 292/308	412	[9]
	Vit A	307/320	410	[22]
A2	Tyr	276	302	[18]
	Tyr	280	305	[60]
A3	Trp	290	340	[6,8,9,17]
A4	AGEs	310–350	380–430	[30]
	Pentosidine	335	385	[65]
	Dityrosine	315	400	[68]
	Vit B6	333	393	[31]
	Vit B6	328	393	[18]
	Vit A	321	412	[9]
A5	AGEs	340–370	420–440	[70]
	AGEs	340–370	420–470	[62]
	AGEs	370	440	[71,72,73]
	Pentodilysine	366	440	[77]
	NADH	360	460	[38]
	NADH	340	450	[22]
	NADH	340	460	[37]
A6	AGEs	340–370	420–440	[70]
	AGEs	350	415	[76]
	pyrropyridine	379	455	[77]
A7	Rb	370	525	[78]
	Rb	370	507	[13]
A8	Rb	380	420/520	[8]
	Rb	370	490/507	[13]
A9	Lumichrome	450	444–479	[20]
A10	Rb	450	525	[78]
	Rb	450	522	[13]
	FADH_2_	450	515	[34]

A1–A10: areas 1 to 10; Vit A: vitamin A or retinol; Vit B6: vitamin B6 or pyridoxine; Tyr: tyrosine; Trp: tryptophan; AGEs: advanced glycation end products; Rb: riboflavin; FADH_2_: coenzyme flavin adenine dinucleotide.

## Data Availability

Not applicable.

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
