# Peer review of "Synchronous Front-Face Fluorescence Spectra: A Review of Milk Fluorophores"

_foods, 2024, doi:10.3390/foods13050812_

Round 1

Reviewer 1 Report

Comments and Suggestions for Authors

Dear authors,

After carefully reviewing the manuscript titled "Synchronous front-face fluorescence spectra: A review of milk fluorophores", I suggested for acceptance with major revision.The manuscript presents detalied all the information on milk fluorescence response. The present review provides a solid foundation for further research and could serve as a central reference. However, this review is still lack of some important highlights to attract readers. Here are some suggestions to help authors improve the paper’s quality.

 Revisions/comments:

Comment 1:  Current version is too concise and it is suggested to extend the content by providing more figures and mechanism depiction to better present.

Comment 2: As a review paper, the number of the present references is not enough (approximately 100), particularly some papers published in recent 5 years should be checked and cited regarding to the fluorescence spectroscopy. It is suggested some of these references:  

Ujjal Bhattacharjee, Danielle Jarashow, Thomas A. Casey, Jacob W. Petrich, and Mark A. Rasmussen.  Using Fluorescence Spectroscopy To Identify Milk from Grass-Fed Dairy Cows and To Monitor Its Photodegradation Journal of Agricultural and Food Chemistry 2018 66 (9), 2168-2173 DOI: 10.1021/acs.jafc.7b05287

Radotić, K.; Stanković, M.; Bartolić, D.; Natić, M. Intrinsic Fluorescence Markers for Food Characteristics, Shelf Life, and Safety Estimation: Advanced Analytical Approach.  Foods 202312, 3023. https://doi.org/10.3390/foods12163023

Comment 3: It is suggest that all references be entered uniformly according to the rules of the journal.

Author Response

Dear Reviewer,

We are sending herewith the manuscript formerly entitled "Synchronous front-face fluorescence spectra: A review of milk fluorophores", Manuscript ID: foods-2853243. As stated in the previous submission, this paper presents an original review that has not been published previously, nor is it under consideration for publication elsewhere.

All comments were analyzed in detail, and our manuscript's revised version incorporates the recommended changes. You will find your comments in blue, followed by our answer in black, which is preceded by an "A." The modified text proposed is presented in italic format. In addition, all changes made to the manuscript have been highlighted so it can be easily reviewed. We have provided a detailed response to each comment, and we are confident that the updated manuscript will meet the expectations of the research community.

We would like to express our sincere appreciation for the detailed review and constructive feedback provided on our manuscript. The suggestions improved the overall quality of our work, and we are grateful for the time and effort spent reviewing it.

Sincerely

Anna Zamora Viladomiu, Ph.D.     

Open Review

Quality of English Language

(x) I am not qualified to assess the quality of English in this paper
( ) English very difficult to understand/incomprehensible
( ) Extensive editing of English language required
( ) Moderate editing of English language required
( ) Minor editing of English language required
( ) English language fine. No issues detected

Is the work a significant contribution to the field?                                                   Starts:4/5

Is the work well organized and comprehensively described?                                  Starts:3/5

Is the work scientifically sound and not misleading?                                              Starts:4/5

Are there appropriate and adequate references to related and previous work?        Starts:3/5

Is the English used correct and readable?                                                                Starts:3/5

Dear authors,

After carefully reviewing the manuscript titled "Synchronous front-face fluorescence spectra: A review of milk fluorophores", I suggested for acceptance with major revision. The manuscript presents detalied all the information on milk fluorescence response. The present review provides a solid foundation for further research and could serve as a central reference. However, this review is still lack of some important highlights to attract readers. Here are some suggestions to help authors improve the paper’s quality.

Revisions/comments:

Comment 1:  Current version is too concise, and it is suggested to extend the content by providing more figures and mechanism depiction to better present.

A: Thank you for your suggestion. We have made significant updates to the content, including the addition of over 1700 words. We provided more information about the fluorophores after the introduction in lines 78-109 to offer a more comprehensive perspective of fluorophores. Furthermore, we appreciate the idea of adding figures. Thus, we have incorporated Figures 1, 2, 3, 5, and 6 into the content.

Comment 2: As a review paper, the number of the present references is not enough (approximately 100), particularly some papers published in recent 5 years should be checked and cited regarding to the fluorescence spectroscopy. It is suggested some of these references:  

  1. Ujjal Bhattacharjee, Danielle Jarashow, Thomas A. Casey, Jacob W. Petrich, and Mark A. Rasmussen.  Using Fluorescence Spectroscopy To Identify Milk from Grass-Fed Dairy Cows and To Monitor Its Photodegradation Journal of Agricultural and Food Chemistry2018 66 (9), 2168-2173 DOI: 10.1021/acs.jafc.7b05287
  2. Radotić, K.; Stanković, M.; Bartolić, D.; Natić, M. Intrinsic Fluorescence Markers for Food Characteristics, Shelf Life, and Safety Estimation: Advanced Analytical Approach.  Foods2023, 12, 3023. https://doi.org/10.3390/foods12163023

A: Thank you for your suggestion.

In this review, the emphasis was on papers that have conducted chemical confirmation of the compounds to ensure that the fluorescent response is indeed from a particular fluorophore. Furthermore, there are studies that have examined the impact of introducing the pure compound to milk samples to verify the fluorescence response or detect any emergence of fluorescence. In fact, this review revealed that only a limited number of investigations have been conducted to chemically verify, or even indirectly verify, the compound in question. Therefore, we have noticed that most of the papers that make an effort to understand their results and explain their own fluorescence response are information recycled. That is why this paper is important because we dug into the literature and tried to show the original papers that put the information first. That is why it seems that we have left some bibliography, but after carefully rechecking it. We can tell that the literature provided is the most relevant. However, the paper that you recommended is a perfect fit, so we have added Bhattacharjee et al. (2018) on lines 392-394: "Bhattacharjee et al. present the spectra of milk and pure riboflavin excited at 420 nm and emitted at 520 nm, showing the third peak of the riboflavin belonging to area 10.", and the second paper (Radotić et al., 2023) helped us to enrich the bibliography on line 65.

Comment 3: It is suggested that all references be entered uniformly according to the rules of the journal.

A: Thanks for your comment. We have checked the bibliography and have corrected references 1, 23, 24, 28, 32, 40 and 68.

Reviewer 2 Report

Comments and Suggestions for Authors

Authors present an original approach based on area identification of synchronous fluorescence maps acquired in front face mode on milk samples. The article is well organized and the topic is of industrial and scientific interest, even if the results presented are not resolutive in indication a unique assignment spectral peak-molecular species given the complexity of the samples. I think some issues need to be solved.

In the introduction is stated (lines 15-17): “However, when applying fluorescence

spectroscopy for any application, there is no need to determine which compound is responsible for each fluorescent response.” In think the need is there, but the complexity of the system and the dependence on several variable parameters makes the unique identification really hard.

It think it could be more clearly stated which kind of milk is being considered for each part of the review, because sometimes seems that only cow milk is being considered, but then some correlations are introduced that refer to other species.

Bibliography could be a little enriched, possibly including papers where synchronous fluorescence or fluorescence have been correlated with other techniques (like NIR spectroscopy, or the use of fluorescence enhancers), and also referring to the kind of multivariate analysis or machine learning approaches can help which methods can be useful for spectral peak identification. Also the possibility to use fluorescence lifetime measurements to reduce ambiguity of concatenation could be cited.

The spectral map shown refers to a single measurement session? Please specify why you take it as example map.

Here some papers that be added to bibliography:

Structural Changes of Milk Components During Acid-Induced Coagulation Kinetics as Studied by Synchronous Fluorescence and Mid-Infrared Spectroscopy

Tahar Boubellouta et al

Rapid discrimination between buffalo and cow milk and detection of adulteration of buffalo milk with cow milk using synchronous fluorescence spectroscopy in combination with multivariate methods

Reviewer 3 Report

Comments and Suggestions for Authors

Paulina et al. summarized the fluorophores appeared when inducing synchronous front-face fluorescence from milk. The fluorescence peaks or areas are related to at least one fluorophore, which will certainly contribute to researchers. Two suggestions are given to improve the quality of this review.

(1)    Is it possible to give some excitation-emission matrices of some pure typical fluorophores? I think this can show the relation of fluorophores and their corresponding fluorescence spectra more vividly.

(2)    Is it possible to add two columns in Table 1, telling what is the detection limit / accuracy achieved, and which method is employed. This would definitely help the researchers to develop more accurate predicting methods of the content in milk in the future.
